# Understanding the Cell’s Response to Chemical Signals: Utilisation of Microfluidic Technology in Studies of Cellular and *Dictyostelium discoideum* Chemotaxis

**DOI:** 10.3390/mi13101737

**Published:** 2022-10-14

**Authors:** Michael Muljadi, Yi-Chen Fu, Chao-Min Cheng

**Affiliations:** 1International Intercollegiate Ph.D. Program, National Tsing Hua University, Hsinchu 30013, Taiwan; 2Institute of Biomedical Engineering, National Tsing Hua University, Hsinchu 300, Taiwan

**Keywords:** chemotaxis, *Dictyostelium discoideum*, diffusion chambers, microfluidics

## Abstract

Cellular chemotaxis has been the subject of a variety of studies due to its relevance in physiological processes, disease pathogenesis, and systems biology, among others. The migration of cells towards a chemical source remains a closely studied topic, with the Boyden chamber being one of the earlier techniques that has successfully studied cell chemotaxis. Despite its success, diffusion chambers such as these presented a number of problems, such as the quantification of many aspects of cell behaviour, the reproducibility of procedures, and measurement accuracy. The advent of microfluidic technology prompted more advanced studies of cell chemotaxis, usually involving the social amoeba *Dictyostelium discoideum* (*D. discoideum*) as a model organism because of its tendency to aggregate towards chemotactic agents and its similarities to higher eukaryotes. Microfluidic technology has made it possible for studies to look at chemotactic properties that would have been difficult to observe using classic diffusion chambers. Its flexibility and its ability to generate consistent concentration gradients remain some of its defining aspects, which will surely lead to an even better understanding of cell migratory behaviour and therefore many of its related biological processes. This paper first dives into a brief introduction of *D. discoideum* as a social organism and classical chemotaxis studies. It then moves to discuss early microfluidic devices, before diving into more recent and advanced microfluidic devices and their use with *D. discoideum*. The paper then closes with brief opinions about research progress in the field and where it will possibly lead in the future.

## 1. Introduction

The directional and preferential movement of cells towards a chemical source—also known as chemotaxis—remains one of the most studied phenomena over the past couple of decades due to its role in biological processes such as embryogenesis, wound healing, and tumour metastasis [1,2]. In physiology, cell chemotaxis also plays a role in the healing and maintenance of tissues [3]. Beyond that, it is one of the key drivers in disease pathology, including rheumatoid arthritis [4] and coronary artery disease [5], which can be attributed to errors in chemotactic action. Another part of this error can also be attributed to cytokine action—cell signalling molecules that regulate immunity—which can be observed in a variety of allergic conditions. For example, there is evidence that eosinophils—a type of white blood cell responsible for combating infections—can be found in asthmatic airways. This suggests an error in cytokine action due to improper cellular signalling, causing a migration of eosinophils to tissues in the airways [6]. This can also be observed in rheumatoid arthritis, where leukocyte chemotaxis and infiltration of the synovium due to cytokine action are major contributors to its pathogenesis [4]. Another example of this phenomenon is in cancer metastasis, where tumour cells spread to other parts of the body, resulting in low rates of survival among patients. This can be attributable to organ-derived chemotactic agents sourced from organs such as the liver, lung, and brain, which have been found to attract tumour cells [7]. This was also found to be linked with cytokine behaviour, particularly in human breast cancer, further highlighting the role of signalling molecules, not just in metastasis but also in tumour cell proliferation [8]. A further understanding of the mechanisms and the role of signalling molecules in chemotaxis is therefore crucial, not just in the understanding of cell biology in general but also in disease pathology, which has implications in human health and therefore quality of life.

Studies of chemotaxis are typically conducted using a variety of models, including prokaryotes such as *E. Coli* [9,10], lymphocytes such as T cells [11], and eukaryotes such as sperm cells [12], among others. *Dictyostelium discoideum* (*D. discoideum*)—a motile soil amoeba approximately 10–20 µm in diameter with six chromosomes [13]—is a leading model for the study of eukaryotic chemotaxis and has been extensively studied due to its simplicity and accessibility. This makes it a powerful tool in the study of cell movement, morphogenesis, and the creation of tissues and organs [14]. As an example, a study in 1989 found that the signal transduction mechanisms in eukaryotes appear to be similar to those of mammalian cells through *D. discoideum* [15]. A more recent paper in 2014 also found that individual signalling molecules, including their architecture, are similar in both mammalian leukocytes and *D. discoideum* [16]. 

*D. discoideum* is known as a social amoeba because of its ability to form aggregates of thousands of cells in response to environmental stimuli and chemical signalling molecules [17]. CAR1—a G-protein-linked receptor in *D. discoideum*—is known to recognize cAMP, a chemoattractant that attracts the microorganism and allows *D. discoideum* to move towards higher concentrations of cAMP [18]. Aggregation is further enforced through cAMP synthesis and secretion by the cells, which further recruits more *D. discoideum* into the cells’ immediate environment [19].

Classical qualitative observations on cell movement towards cAMP mostly rely on the micropipette [20], where cells are seen to lean and move towards stationarily emitted cAMP from the tip. These tend to be qualitative observations often and do not have clearly defined gradients. Moreover, the concentrations of these assays can change over time due to the diffusion of the chemoattractant. In other typical chemotaxis assays, concentration profiles of the chemoattractant are typically generated via diffusion from a source to a sink. One of the earlier studies of this was in 1962 and looked at the movement of leukocytes in response to the presence of tuberculin [21]. The design—which would later be known as the Boyden chamber—used a Perspex chamber separated into two compartments, with a filter through which leucocyte cells could only go through under active migration towards the chemoattractant tuberculin. Through this, Boyden found that cells responded to a difference in concentration through active migration to a medium of higher concentration. Boyden’s chamber would later become a setup used to study leukocyte and fibroblast [22] chemotaxis. 

A study in 1991 by Zicha and colleagues then looked at the predictability of gradient formation by incorporating Helber bacteria counting chambers, allowing for direct observation and a more precise quantification of the cells [23]. This was a modification from another well-known chamber known as the Zigmond chamber [24], which was used to study chemotaxis in leukocytes. The concentration difference was set up between a circular outer chamber and an inner chamber. The two wells were first filled with control medium, after which they were covered by a 24 × 24 mm coverslip with adhering cells. A gap was initially left open, allowing a syringe to deliver the test solution into the outer well. The gap was later sealed using a hot wax mixture. The test solution then travelled to the inner chamber via diffusion, and cell movement was observed via microscopy. Using a fluorescent dye, Zicha and colleagues found that they were able to predict the concentration gradient of the chamber using mathematical models of diffusion theory. They believed that their design could be advantageous in studies where precise knowledge of the gradient is required.

Boyden and Zicha’s studies (Figure 1A) highlighted significant implications. The Boyden study found that cells tend to travel up a concentration gradient independent of random movement. The study was also able to provide the means to make quantitative estimates of the migrated cells through microscopy, but there was still room for improvements to better understand behaviours, such as the identification of the chemotactic factors that influence the cells. This required a more accurate method of quantification of both the chemotactic gradient as well as the cells. The chamber design was also such that it was impossible to observe the cells as they were moving, and this made it difficult to conduct further analyses on chemotactic behaviour. As for Zicha’s study design (Figure 1B), although they found that concentration profiles could be reliably predicted through diffusion theory, the process was still slow, tedious, and unstable, as small changes in the chamber dimensions or procedure could result in thermal expansions, mechanical creep, and surges in the medium. Therefore, based on the limitations of these two studies, there was a need for a technique that could allow a more accurate quantification of the chemotactic gradient and the cell, where consistent parameters could be produced with minimal effort, and that was where microfluidics started playing a significant role in chemotaxis studies.

## 2. The Advent of Microfluidics in Chemotaxis Studies

In 1989, Fisher and colleagues conducted a quantitative analysis of *D. discoideum* chemotaxis using an automated image processing system in a chemotaxis chamber (Figure 2A) [25]. Similar to the Zigmond design, the chamber contained a viewing window from which the cell motility was allowed to be observed. A stable linear chemical gradient was formed using a pump that released both chemoattractant and buffer solutions. Based on their observations of chemotaxis dependence on gradient steepness, the findings suggested that *D. discoideum* respond to local increases in chemoattractant concentration and adjust their movements accordingly. This further reinforced previous findings on cellular locomotion up a linear gradient but still required a more comprehensive technique to further develop the stability of the concentration profile. Cells can release its own chemicals in response to external stimuli. In the case of *D. discoideum*, the chemoattractant cAMP is gradually released as it responds to it in the external environment [19]. None of the chambers by Boyden, Zigmond, Zicha, or Fisher took into account the cells’ own influence on their chemical environment. Released chemicals remained in the chambers and were not gradually removed upon secretion. This could have affected the interpretation of cell behaviour and therefore remained one of the disadvantages of these studies.

Microfluidic technology allows for the generation of more precise concentration profiles and the ability to maintain said concentration profiles over long periods of time. One of the earlier studies that carried out these procedures was in 2000 by Jeon, Whitesides, and colleagues of the University of Harvard (Figure 2B). Jeon fabricated a gradient generator using microfluidics created by soft lithography and rapid prototyping using poly(dimethylsiloxane) (PDMS) [26]. The structure involved three inlets and one outlet with an interconnected network of channels in between. The mixture and buffer were introduced from any or all of the three inlets, where they travelled via a laminar flow with a low Reynolds number through the multitude of channels towards the outlet. The channels were combined into a single channel towards the end, where the solution established a gradient perpendicular to the direction of the flow. The gradient generation involved a controlled diffusive mixing of fluids within the network of microchannels and recombining them via laminar flow to form a precisely manipulated concentration profile. The width and resolution of the gradient profile were controlled by the channel width and the total number of branches within the network. Moreover, microfluidics is able to remove chemicals released by the cells themselves, allowing for a more accurate interpretation of observed chemotactic behaviour. They found that they were able to reliably generate gradients of different sizes, resolutions, and shapes using this technique.

Various gradient studies have taken advantage of Jeon’s 2000 microfluidic model to generate their linear gradient profiles. One these was a follow-up study conducted again by Jeon and colleagues in 2002, where they looked at the movement of human neutrophils in response to changes in the interleukin-8 (IL-8) concentration. The use of microfluidics-generated concentration profiles also allowed them to look at more complex migratory behaviour, including directional sensing [27]. Using the 2000 study’s microfluidic gradient generator, Jeon and colleagues randomly seeded neutrophils throughout the observation channel and found that the neutrophils were able to change directions if presented with multiple competing chemoattractants. Such an observation would not have been possible under Zicha’s inner and outer well design [23].

Another study in 2004 made use of the same technique in a cancer metastasis application. Wang and colleagues looked at the chemotactic behaviour of breast cancer cells (MDA-MB-231) in epidermal growth factor (EGF) gradients [28]. Microchannels were also fabricated in poly(dimethylsiloxane) (PDMS) using soft lithography, and the two inlets was filled with medium and EGF-filled medium. From this setup, they were also able to observe cells moving up and away from a higher concentration gradient and suggested that a threshold exists where the solution can induce an effective cell chemotaxis. These findings—consistent with previous studies—suggested that microfluidics allows for the precise and robust manipulation of parameters for the study of cell chemotactic behaviour, which became the basis for microfluidic chemotaxis assays.

## 3. Microfluidics in Studies of *Dictyostelium discoideum* Chemotaxis

Microfluidics allowed for more advanced studies (Figure 3) that would have been difficult to execute using conventional techniques such as: threshold determination for directed motion [29], cell movement in alternating gradients [30], the construction of a probabilistic model for eukaryotic chemotaxis [31], and barotactic guidance cues in *D. discoideum* [32], among many other studies.

### 3.1. Directed Motion Threshold

Progress in the field prompted Song and colleagues to use microfluidics in the study of *D. discoideum* chemotaxis using linear gradients of cAMP in 2006 [29]. In particular, based on earlier findings on the existence of a threshold, they took advantage of the ability of microfluidics to generate controlled linear concentration gradients to observe the *D. discoideum* response. Apart from the microfluidic device’s ability to maintain stable gradients, it was also able to continuously eliminate the excess chemoattractant excreted by the cells themselves, which could have affected the analyses. A more precise control of linear concentration gradients of cAMP made it possible for Song and colleagues to determine a threshold in which the organism is able to move and maintain directionality and motility. 

For the microfluidic device, Song and colleagues used a modified pyramidal network based on the study by Jeon and colleagues in 2000 [26]. The structure used a cascade mixing procedure to combine cAMP with a developmental buffer, where solutions of minimal and maximal cAMP concentrations were introduced through two inlets (Figure 3A). As the device continued to branch into multiple channels, the solutions continued to mix via diffusion with their neighbouring channels. This resulted in multiple channels holding different cAMP concentrations in a stepwise fashion. These channels merged at the end into a single channel to produce a linear gradient that was perpendicular to the direction of the flow. A steady flow was maintained using a syringe pumping system at a flow speed of 650 μm/s, ensuring that the concentration gradient was maintained, cell by-products were removed, and at the same time preventing the cells from being washed away. Cell coordinates were measured in areas where the gradient was constant.

Song and colleagues made two observations: motion in the absence of a gradient and motion in the presence of linear gradients. They found that in the absence of chemoattractant the cells exhibited random motion with no directional preference, suggesting that the setup did not have a significant impact on cell motion. In instances where linear gradients existed, directional responses were observed with a lower gradient threshold value of 10^−^^3^ nM/μm. The distribution of velocity in the x direction was found to be similar to the case of the absence of gradient, while the distribution in the y direction highly favoured the direction of gradient. They also found that at gradient values above 3.3 nM/μm the behaviour of cells returned to subthreshold values. These findings suggested that directed motion exists within a threshold of linear gradients and that, in the particular case studied by Song and colleagues, they lie between 10^−^^3^ and 3.3 nM/μm. Moreover, the use of microfluidics to wash away cell secretions allowed them to suggest that random motion does not solely depend on intercellular signalling but also depends on some intrinsic cellular mechanism that is yet to be discovered.

### 3.2. Directional Sensing in Alternating Gradient Fields

The existence of a threshold for directed motion opened many doors for observations of *D. discoideum*. In 2011, Meier and colleagues studied the chemotactic behaviour of *D. discoideum* in alternating gradient fields using microfluidic technology [30]. Meier and colleagues took advantage of the threshold to study the chemotactic responses of single cells to modifications in the shapes and positions of gradients as a function of switching frequencies. *D. discoideum* was used because of its rapid chemotactic characteristics, allowing them to test the experimental limits of rapid changes in alternating gradients (Figure 3B).

Meier designed a microfluidic function generator consisting of a double T-junction chamber and three inlets with variable pressures. The three inlets included the central inlet in the middle and two side inlets, one on each side. The central flow contained no stimulants, while the side flows introduced the chemotactic agents in an alternating fashion. The introduction of the chemoattractant on the SF1 side prompted directed cell migration in that direction, while an introduction on the SF2 side prompted a switch in polarization, forcing the cells to migrate in the opposite direction. This experimental setup allowed them to measure intracellular response dynamics and reorientation timescales. They found that at a gradient switching rate of 0.02 Hz *D. discoideum* migrated towards the new direction of the chemotactic stimulus. However, at a faster rate of 0.1 Hz, the cells were trapped in a chemotactic state, suggesting that their intracellular feedback is incapable of responding to rapid changes in external stimuli in directed migration. This also allowed them to observe an adaptation time of approximately 3 minutes, where the *D. discoideum* could rearrange their actin polymerisation activity following a switch in gradient direction before migrating in the reverse direction. Apart from these experiments, microfluidics also allowed Meier and colleagues to study the effects of cell starvation on polarisation in changing gradient steepness conditions.

### 3.3. Barotactic Guidance Cues in Confined Environments

In 2020, Belotti and colleagues looked at the directional movement of *D. discoideum* towards the path of least hydraulic resistance, also known as barotactic behaviour [32]. The study looked at cell migratory behaviour in an environment that more closely resembles those in tissues and other complex environments. Studies have looked at cell chemotaxis on planar surfaces but not so much in confined environments. It was argued that cells tend to take advantage of both chemical and mechanical cues for navigation within complex environments. It has been shown that cells such as human promyelocytic leukaemia cells are sensitive to changes in hydraulic resistance, moving towards the path of least resistance in the absence and in the presence of chemotactic agents [33]. Belotti and colleagues attempted to further understand barotactic behaviour through microfluidic technology, allowing them to observe cell behaviour when presented with two choices: increasing chemical gradient and lower hydraulic resistance (Figure 3C). To differentiate said mechanical and chemical driving forces, Belotti and colleagues designed a microfluidic device with microchannels that decoupled hydraulic and chemical stimuli. At the point of bifurcation, cells were subjected to a dual choice where one arm corresponded to an increasing chemical stimulus while the other corresponded to a lower hydraulic resistance. Where the arm was connected to a chemoattractant source, the microfluidic channel was designed in such a way that hydraulic resistance was a hundred times higher when compared to the other arm. Belotti and colleagues therefore achieved the discrimination of mechanical and chemical driving forces through microfluidic channel design.

They found that cells tend to move up a chemical gradient regardless of high amounts of hydraulic pressure. They were also able to perform a separate analysis confirming the tendency of cells to migrate towards the path of least resistance through changes in the hydraulic resistance ratio between two bifurcating arms. Fast-moving cells tended to split and protrude upon reaching a bifurcation before ultimately choosing the right channel depending on the experimental conditions. Despite plugging the channel, they theorised that cells were still able to move because of a difference in the cAMPS concentration between the front of the cell and the back of the cell. The extension–retraction process was then governed by changes in actin–myosin cytoskeleton organisation. They also found that higher cAMPS concentrations resulted in faster-moving cells, regardless of hydraulic resistance. These further highlighted the cell’s directional bias towards higher concentrations of chemoattractant, suggesting that cellular migration is driven by chemotactic signalling and not hydraulic cues. Such analyses on guidance cues and cellular decision-making processes would not have been possible with a micropipette [20], and proper quantification would have been difficult with the classic diffusion chambers [22,24].

### 3.4. High Sensitivity to Low Chemoattractant Concentrations

In 2021, Ohtsuka and colleagues studied the heterogeneity of chemotaxis responses in a homogeneous *D. discoideum* cell population under the same chemoattractant gradient [34]. The study employed 800 cell-sized microfluidic channels, each 4 µm in width, 5 µm in height, and 100 µm in length, to expose 500–1000 individual cells to the same chemical gradient. The objective was to study the chemotaxis range of a population of *D. discoideum*, their chemotaxis efficiency, and the dependence of chemotaxis on cAMP concentration. The chemotaxis efficiency was defined as the percentage of cells in the population that exhibited chemotaxis in a given cAMP concentration range. In the absence of cAMP, they found that 14.9 ± 3.4% of the cells reached the right half region of the microfluidic channel within 60 minutes by spontaneous migration. This was their control condition. At a range of 10–12 to 10–11 M cAMP, they found that significantly more cells migrated toward the chemoattractant compared to the control, suggesting that there was evidence of a subpopulation of *D. discoideum* cells that responded to even very low concentrations of cAMP at 10–11 M. Based on these results, they determined that the chemotaxis range of *D. discoideum* ranged from 0.1 pM to 10 µM cAMP.

Microfluidics enabled the construction of microchannels the size of single *D. discoideum* cells, allowing for the study of subpopulations that respond to different ranges of cAMP concentrations. Further advances in microfluidics enabled Ohtsuka and colleagues to further elaborate on the findings of Fisher’s 1989 study, where the chemotaxis range of *D. discoideum* was found to be approximately 10–10 to 10–6 M [25]. This is because the cells were found to have the highest chemotaxis efficiency at a cAMP concentration of around 10–7 M, which was perhaps more prominent in Fisher’s 1989 study. The subpopulation of *D. discoideum* that is sensitive to very low concentrations of cAMP was thought to be the one promoting the onset of aggregation among the majority of the population. The findings of Ohtsuka and colleagues were important in helping future studies characterise the transition of the social amoeba from single cells to multicellularity.

### 3.5. Spatial Confinement and Migratory Properties

As a follow-up and with the same rationale as their previous study, Belotti and colleagues looked again at the migratory properties of *D. discoideum* in confined environments to simulate in vivo conditions [35] using microfluidic technology. The actomyosin cytoskeleton response to chemotactic cues was studied in a ladder-like microfluidic configuration where the chemoattractant concentration was adjusted through the volume of media in the loading reservoirs. The effects of spatial confinement on the migration velocity were studied by controlling the widths of the microfluidic migration channels at both constant and progressively decreasing widths. Apart from observing a significantly lower migration velocity at narrower channel widths, they also found two different modes of motion: pseudopod-based migration and blebbing. In the pseudopod-based migration mode, they found that the protrusions tended to be irregular and that F-actin was involved, whereas F-actin was absent in the early stages of formation and the protrusions were faster in blebbing. *D. discoideum*’s tendency to protrude blebs was also attributed to the decreasing size of the microfluidic channel, and the quick shifting of entire cell bodies was observed. Therefore, there is evidence that *D. discoideum*’s migratory properties are dependent on the degree of spatial confinement, which also determines the mode of migration, which depends on the amount of F-actin at its leading edge.

## 4. Discussion

The emergence of microfluidics as a robust analytical tool for a wide array of applications in the late 20th and early 21st century have helped to further progress scientific research in a variety of fields. A short summary of the techniques used in studies of cellular chemotaxis prior to the emergence of microfluidics is presented in Table 1, while Table 2 presents a glimpse of the progress in understanding the phenomenon through microfluidic technology and the model organism *D. discoideum*.

The absence of clearly defined gradients in qualitative chemotaxis observations prompted more well-controlled techniques. One of the earlier ones was the Boyden chamber back in 1962, which looked at the leukocyte chemotactic response due to the presence of tuberculin [16]. A modified version of the chamber was later used by Postlethwaite and colleagues in 1976 [36] as a means to further understand the mechanism by which fibroblasts are attracted to sites of inflammation, highlighting the role of chemotaxis in wound healing [3]. The Boyden chamber allowed them to observe the fibroblasts’ chemotactic response to T-lymphocyte-derived factor in vitro, but its limitations meant that more specific observations, such as the fibroblast response to linear gradients of tuberculin, were not possible.

Zigmond specifically mentioned this as part of his study in 1974, when observing the polymorphonuclear leukocyte response to chemotactic gradients [18]. Zigmond also mentioned the need to understand the time and concentration thresholds of cell stimulation as well as their internal means of comparing different stimuli, which would have been difficult to observe through classic diffusion chambers. Microfluidic technology was able to achieve these goals; Song and colleagues observed the directed motion threshold of *D. discoideum* in linear gradients of cAMP [23] using a modified version of a pyramidal network designed by Jeon and colleagues [20]. Microfluidics also allowed observations of directional cues in the face of differing stimuli, as was evident in a 2020 study by Belotti and colleagues, which looked at the barotactic behaviour of *D. discoideum* when faced with differing chemoattractant concentrations and hydraulic pressures [26]. These two studies directly confronted issues posed by Zigmond, highlighting the strengths of microfluidics as an analytical tool. 

Studies using classical diffusion chambers did not end with Zigmond; Fisher and colleagues attempted to modify Zigmond’s chamber, forming linear chemical gradients using a pump that released a chemoattractant and a buffer solution [19]. Fisher’s methodology was one of the closest to microfluidics in terms of its attempt at controlled diffusion via a pump. Although the stability of the concentration profile could have been improved, they found that the cells were able to adjust their motility in response to changes in external stimuli, encouraging further understanding of more complex cellular behaviours such as directional sensing. In 2002, using microfluidic technology, Jeon and colleagues found that neutrophils had directional preferences when presented with competing chemoattractants [21]. This was also confronted by Meier and colleagues in 2011, again using microfluidics to study *D. discoideum* reorientation in the presence of alternating gradient fields [24]. This allowed them to observe the cellular adaptation time following a switch in the gradient direction. Similar to the Belotti study from 2020, these studies took advantage of the robust capabilities of microfluidics to confront one of Zigmond’s proposed issues on cellular directional cues when faced with differences in external stimuli. Similarly, this can also be observed in more recent studies of microfluidics, even those not studying *D. discoideum*, such as bacterial chemotaxis and cancer cell chemotaxis. Salek and colleagues, for example, exposed clonal *Escherichia coli* (*E. coli*) to T junctions with differing chemical gradients in 2019 [37]. Through this study, they were able to infer nongenetic heterogeneity in *E. coli*’s chemotactic behavioural decision making. This is similar to Belotti’s 2020 study on *D. discoideum*’s chemotactic behaviour when presented with competing stimuli [32]. Belotti’s 2021 observation on *D. discoideum’s* behaviour in a simulated confined environment using microfluidics can also be observed in another study by Chen and colleagues in 2015. Chen took advantage of microfluidic technology and modified migration channels that resembled lymphatic capillaries to study cancer cell chemotaxis in confining spaces [38]. Both Salek’s and Chen’s studies, parallel to Belotti’s studies, showed that there is a pattern to the progress in chemotaxis studies, regardless of organism, and that they are possible because of microfluidic technology.

Based on the literature, there has been a shift in focus in chemotaxis studies from the early 1960’s to the early 2020’s due to the introduction of microfluidics. More recent studies have been able to look at phenomena that would otherwise have been hard to observe without microfluidic technology. Prior to the year 2000, studies focused more on the generation of a chemoattractant gradient, while studies after Jeon’s introduction of a pyramidal network of microfluidic channels focused more on cell mechanisms and decision making in response to a variety of chemotactic cues and changes in the cell’s microenvironment. Boyden, Zigmond, Fisher, and Zicha all mentioned not being able to remove the secreted chemical by-products of cells, while John not only provided a solution to this but also provided a solution to the reliable generation of concentration gradients. From then on, studies using both *D. discoideum* and microfluidics provided a better understanding of cell migratory behaviour, which can potentially help in understanding cell behaviour in crucial areas such as cancer. Fisher’s 1989 study, for example, found that *D. discoideum* had a chemotaxis range of 10–10 to 10–6 M under a simplified microfluidic system. Ohtsuka’s more recent 2021 study, however, found that, based on their microfluidic channel design, *D. discoideum* had the highest chemotaxis efficiency at 10–7 M cAMP, which was suggested as the reason why Fisher postulated a chemotaxis efficiency of 10–10 to 10–6 M. Moreover, as microfluidics allowed them to construct single-cell-sized channels, they also found a subpopulation of cells that even responded to very low cAMP concentrations of 10–11 M. The existence of this very sensitive subpopulation suggested that these cells might be the ones responsible for the aggregation of the larger population. The two other Belotti studies also focused on the effects of microenvironmental parameters on cell decision making and migratory behaviour, which were made possible by microfluidic technology. 

One possible future direction is the incorporation of elements of these different studies to study more advanced processes. Combining Ohtsuka’s study and two of the Belotti studies, for example, can provide insights into how subpopulations of *D. discoideum* cells respond to different environmental parameters such as barotactic guidance cues and varying channel sizes. One example of this would be to construct multiple uniform microfluidic channels—as was incorporated in Ohtsuka’s [34] study—that were designed to have Belotti’s separation of barotactic and chemotactic cues [32] as well as changing channel sizes [35] to study these decision-making mechanisms at the population level. It is possible that a very sensitive subpopulation of Ohtsuka’s *D. discoideum* respond differently in Belotti’s channels, which could provide insight into how cancer cells proliferate in the body.

It has also been found that the mediation of chemotaxis is frequently mutated in cancer [39]. A better understanding of the chemotactic behaviour of cells could therefore allow for the building of more predictive models of chemotaxis in cancer [40]. This is also present in studies that look at the biochemical pathways that underpin chemotaxis, which also utilise microfluidics in their methodology. An example of one such study was a 2019 study by Senoo and colleagues, who looked at the activation of mTORC2 kinase towards AKT to regulate cell migration in *D. discoideum*, the mechanisms of which are prominent in cancer cell proliferation and metastasis [41]. These can open doors for other studies into targeted cancer therapies through predictions of the direction of cancerous growth as well as the reprogramming of chemotactic pathways in favour of tumour cell dissemination, among others. This is also especially true because *D. discoideum* itself has been used in studies to identify at least 18 chemokine receptors for at least 50 different chemokines that control chemotaxis [42]. Coupling that with the many potential studies that can be designed using microfluidics, this technology can bring hope to the millions diagnosed every year.

## 5. Conclusions

All of the studies by Song [29], Meier [30], Belotti [32], and Ohtsuka [34], among others, have studied *D. discoideum* as a model organism through the help of microfluidic technology. Its aggregation characteristic in response to chemical signalling molecules [12] could be used to study other cellular behaviours in response to chemotactic agents. The improper cellular signalling of eosinophils was found to contribute to asthma [6], and cytokines were found to play a role in tumour cell proliferation [8]. Parallels could be drawn between the behaviour of these types of cells and *D. discoideum*, meaning that continuous effort in more advanced studies of *D. discoideum* chemotaxis is crucial for a deeper understanding of disease pathogenesis and cancer metastasis. Not only that, with further advances in microfluidic technology and the combination of already established techniques, the use of model organisms may not be confined to just *D. discoideum* but could include a whole array of microorganisms, especially with advances in smaller and more precise devices in recent years compared to the older classical devices prior to the advent of microfluidics.

Given the number of studies that have relied on microfluidics to look at cell chemotaxis, future studies could most definitely rely on the endless possibilities of microfluidic technology to study the deeper intricacies of chemotactic phenomena and therefore the biological processes that have been—and are yet to be—discovered.

## Figures and Tables

**Figure 1 micromachines-13-01737-f001:**
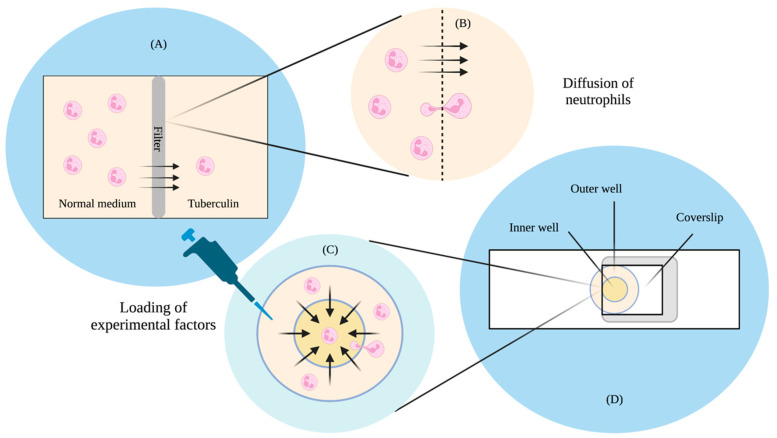
Simplistic adapted diagram highlighting: (**A**) the chemotaxis of neutrophils from the normal medium to tuberculin through a filter in the Boyden chamber; (**B**) the diffusion of neutrophils in the Boyden chamber; (**C**) the loading of experimental factors in the Zicha chamber; and (**D**) the Zicha chamber incorporating the Helber bacteria counting chamber and the diffusion of cells from an outer well to an inner well under a coverslip. Arrows indicate the direction of cell flow. Created with BioRender.com (https://app.biorender.com/). Last accessed on 29 September 2022.

**Figure 2 micromachines-13-01737-f002:**
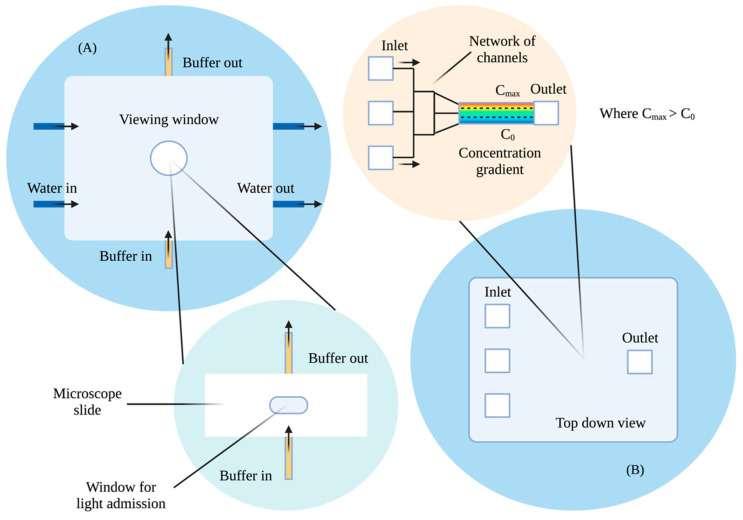
Simplistic adapted diagram highlighting: (**A**) Fisher’s chemotaxis chamber for the quantitative analysis of D. discoideum, with a microscope slide at the bottom and a viewing window at the top for image analysis and (**B**) a top-down view of Jeon and colleagues’ microfluidic gradient generator with an embedded network of channels inside. Arrows indicate the direction of fluid flow. Created with BioRender.com (https://app.biorender.com/). Last accessed on 13 October 2022.

**Figure 3 micromachines-13-01737-f003:**
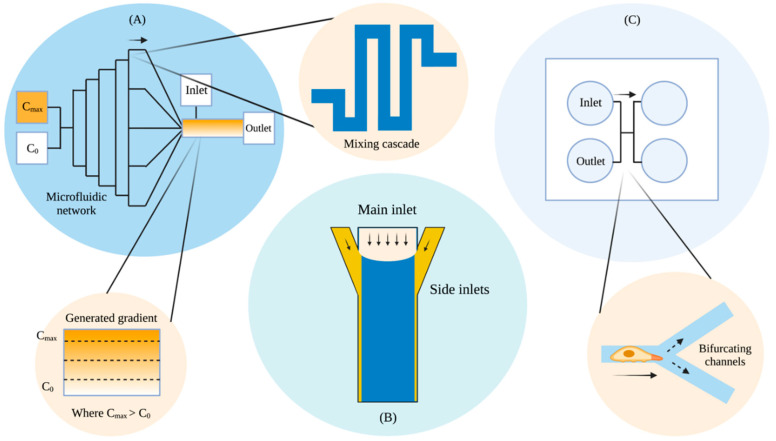
Simplistic adapted diagram highlighting: (**A**) Song’s generation of a linear concentration gradient through a pyramidal microfluidic network of mixing cascades for the study of *D. discoideum* chemotaxis; (**B**) Meier’s tuneable microfluidic function generator through controlled flow for the study of directed cell migration; and (**C**) a top-down view of Belotti’s microfluidic chip with a decoupling of the barotactic and chemotactic channels for the study of *D. discoideum*’s directional decision making. Arrows indicate the direction of fluid flow and cell movement, where applicable. Created with BioRender.com (https://app.biorender.com/). Last accessed on 19 September 2022.

**Table 1 micromachines-13-01737-t001:** Studies on cell chemotaxis using classical diffusion chambers.

Study	Defining Features and Observations	Disadvantages
Boyden chamber (1962)	Used to study movement of cells in response to the presence of tuberculin Three sets of three chambers with two compartments each to generate concentration difference Found that cells tended to migrate to regions of higher tuberculin concentration	Difficult to quantify and track differences in concentration Difficult to directly observe and quantify cells, especially when moving Did not allow for the removal of secreted chemicals
Zigmond chamber (1974)	Used to study leukocyte chemotaxis Concentration gradient established between a circular outer chamber and inner chamber Test solution travelled to the inner chamber via diffusion Incorporated Helber bacteria counting chambers, allowing for the observation of cell movement via microscopy	Accurate concentration profiles remained difficult to quantify and represent No way of removing secreted chemicals
Fisher chamber (1989)	Used an image processing tool to study *D. discoideum* chemotaxis through a chamber similar to that of Zigmond Found that the cells adjusted their movements based on local differences in chemoattractant concentration	Concentration gradient stability still needed improvement Still no way of removing chemicals secreted by the cells, potentially impacting results
Zicha chamber (1991)	A modified version of the Zigmond chamber through the use of a fluorescent dye	Concentration profiles remained unstable and were established slowly
	Successfully predicted concentration gradient through mathematical models of diffusion theory	

**Table 2 micromachines-13-01737-t002:** Studies on *Dictyostelium discoideum* chemotaxis that took advantage of microfluidic technology as well as their implications.

Study	Defining Features	Main Observations	Implications
Directed motion threshold, Song (2006)	Used a modified pyramidal microfluidic network by Jeon (2000) to study *D. discoideum* directed motion threshold Stable gradient and removal of excess chemoattractant	*D. discoideum* exhibited random motion with no directional preference in the absence of a chemoattractant. *D. discoideum* exhibited directed motion at a threshold of 10^−^^3^ and 3.3 nM/μm in the presence of a chemoattractant.	Random motion could potentially be a resultant of some form of intrinsic cellular mechanism.
Directional sensing in alternating gradient fields, Meier (2011)	Used a microfluidic function generator consisting of a double T-junction chamber and three inlets Studied chemotactic response of *D. discoideum* when faced with alternating gradients	*D. discoideum* migrated towards a new direction of chemotactic stimulus at a gradient switching rate of 0.02 Hz. *D. discoideum* was trapped in a chemotactic state when faced with a gradient switching rate of 0.01 Hz.	Different types of cells could have their own responses to rapid changes in external stimuli.
Barotactic guidance cues, Belotti (2020)	Studied *D. discoideum* behaviour towards the path of least hydraulic resistance Studied behaviour when presented with increasing chemical gradient and lower hydraulic resistance	*D. discoideum* moved up a chemical gradient regardless of high amounts of hydraulic pressure. Slow-moving cells tended to move directly into the correct chamber when faced with multiple competing pathways.	Cellular migration could be driven more by chemotactic signalling than hydraulic cues.
Sensitivity to chemoattractant, Ohtsuka (2021)	Studied the chemotaxis range of *D. discoideum* populations Studied the effect of different cAMP concentrations on chemotaxis efficiency	*D. discoideum* has the highest chemotaxis efficiency at a cAMP concentration of around 10^−^^7^ M. There is a subpopulation of *D. discoideum* that responds to very low concentrations of cAMP of 10^−^^11^ M.	The very sensitive subpopulation of *D. discoideum* could play a major role in the aggregation of single cells into multicellularity.
Spatial confinement on migratory properties, Belotti (2021)	Studied *D. discoideum* cell migratory behaviour due to spatial confinement Studied the effect of spatial confinement on the cell’s mode of migration Studied the effect of the mode of migration and focal adhesions on the cells	*D. discoideum* has a tendency to protrude blebs at decreasing channel sizes. F-actin is involved in pseudopod-based migrations. Quick shifting of entire cell bodies was observed during blebbing.	The microenvironment’s dimensionality has an impact on cell migration.

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
