# Peer review of "Understanding the Cell’s Response to Chemical Signals: Utilisation of Microfluidic Technology in Studies of Cellular and Dictyostelium discoideum Chemotaxis"

_micromachines, 2022, doi:10.3390/mi13101737_

Round 1

Reviewer 1 Report (New Reviewer)

1.       The authors wrote a review article about the effect of chemical signals in the chemotaxis of an amoeba D. discoideum. The topic is interesting, however, it would be more interesting if they investigate the effect of chemical signals on mechano-transduction like mTOR pathway.

2.       The scientific name of organisms such as Dictyostelium discoideum must be italic in the whole of the text including the title.

3.       Figure 1 should be sub-named (A)-(D) since it has four sections. The sub-section left–down ( loading of experimental factors) should become larger with higher resolution.

4.       All of the references are too old. Almost the references belong to 1974-2007. If there is no recent reference, it means this topic is expired.  This is very important for review. Referencing should be updated.

5.       In the microfluidic platform, how do you discriminate between driving force and chemical force? You cannot make the pressure to 0 since you have flow, then you have a driving force. How do you ensure that this force does not affect chemotaxis?

6.       There is a lot of simulation result that you can use to enrich your article. 

Author Response

Reviewer 2 Report (New Reviewer)

The manuscript entitled "Understanding the cell’s response to chemical signals: utilisation of microfluidic technology in studies of cellular and Dictyostelium discoideum chemotaxis " by Muljadi et al. showed research progress on the topic of the migration of cells towards a chemical source. The main elements of this manuscript include: chemotaxis in Dictyostelium and other cells; the advent of microfluidics on chemotaxis studies; microfluidics in studies of Dictyostelium discoideum chemotaxis. Overall, this review needs a significant improvement. The following issues should be addressed before it can be accepted for publication.

1.      The manuscript presents a list of the research methods and results, but  provides very few own thoughts from the literature.

2.      It is better to use more schematic figures in the review paper.

3.      Literature number is not enough and some references are old.

4.      Reference 1 cited by the authors does not appear in the main body of this manuscript

Author Response

Reviewer 3 Report (New Reviewer)

The authors give full survey on chemotaxis behavior of cells and demonstrate chemotaxis effect using microfluidic device.

Overall, the review article is well oragnized with sufficient references. The article can be accepted for publication after the following points are reflected.

1. In the abstract, the authors should mention the overall construction of the review article and also mentioning how many references are used.

2. Please delete dots after "Figure". For example, "Figure 1.A." should be "Figure 1A". Please correct all the releated expressions throughout the manuscript.

3. References should be cited using bracket [ ] rather than parenthesis (  ).

4. The "Microfluidics" in the subtitle "3. The advent of Microfluidics on chemotaxis studies" should start with lower case "m" for "Microfluidics".

5. For Fig. 3, if possible, please provide a few more examples of using microfluidic device.

Round 2

Reviewer 1 Report (New Reviewer)

The authors answered my questions. I accept this manuscript, however, it will be more interesting if the authors explained the mTOR pathway's function in the regulation of  chemotaxix  in Dictyostelium discoideum

Author Response

Dear Reviewer,

Thank you very much for your recommendation for acceptance.

Reviewer 2 Report (New Reviewer)

The authors have revised the manuscript very carefully and I am satisfied with the current form.

Author Response

Dear Reviewer,

Thank you very much for your recommendation for acceptance.

This manuscript is a resubmission of an earlier submission. The following is a list of the peer review reports and author responses from that submission.

Round 1

Reviewer 1 Report

The review presented by Muljadi and coauthors summarized the historical progress of cellular chemotaxis, with a focus on dictyostelium chemotaxis based on microfluidic technologies. Along with a clear introduction on why microfluidics is beneficial in this regard, multiple advanced studies such as directed motion threshold, directional sensing in alternating gradient fields and deterministic and stochastic modelling for directed motion have been clearly described. In addition, discussions were included. Overall, the paper is well-written, easy to follow and should be published, yet some minor comments should be considered before its publication.

  • The quality of the figures does not match the quality of the manuscript, please consider using high-quality figures and possibly add more subfigures in a figure to describe similar studies, in case some readers tend to read figures instead of the main text.
  • Not sure why Tables were categorized in case studies chronologically, would it be possible to include similar studies in the tables as well and categorize them in generalized types?
  • Though discussions and a brief conclusion were included, it would be useful to add a separate conclusion section.
  • Please add author contribution information.

Reviewer 2 Report

The authors review focuses on the use of microfluidic technology to study chemotaxis. The chosen cells of interests are D. discoideum and authors have reviewed some of the related literature. While the topic is relevant for a broader biomedical audience, I find this submission lacking rigor and quality to be considered as a publication. 

Major-
1) Figures- Instead of using copyright permission granted figures, authors have reproduced the figures using some web application. This is clearly not acceptable for any scientific peer reviewed journal.
Please review copyright permission processes. 

2) Overall scope and organization- The paper really lacks an outline of scope in the introduction. How many papers were reviewed and how this review is different from the ones that are already published. There is no mention of existing review papers on microfluidics and chemotaxis. An eg- https://doi.org/10.1016/j.tcb.2011.05.002

Additionally, there is no emphasis on the "microfluidic technology" description and several details are missing. How are these devices made? What are the limitations? Details of experimental set up? As a reader, a lot of this information is lacking. 

3) Redundancy- There are so many paragraphs that are verbose and don't really convey any meaningful message. For example, while authors at length describe the previous methods for chemotaxis assays-Boyden chamer, there is not enough explanation for microfluidic devices. 

Many times text repeats itself. For example below- 
Line 212-213 Mentions Microfluidics ability to remove waste
Line 167-168 again mentions microfluidics ability to remove waste

4) Quality- The paper lacks overall scientific quality for a review paper. I would encourage authors to read some published review articles to strengthen this paper.